# Characteristics of Sound Attenuation by Individual and Multiple Fishes

Hansoo Kim [1], Sungho Cho [1], Jee Woong Choi [2] and Donhyug Kang [1],*

[1] Marine Domain and Security Research Department, Korea Institute of Ocean Science & Technology (KIOST), Busan 49111, Republic of Korea; hskim@kiost.ac.kr (H.K.)

[2] Department of Marine Science and Convergence Engineering, Hanyang University-ERICA, Ansan 15588, Republic of Korea

* Correspondence: dhkang@kiost.ac.kr; Tel.: +82-51-664-3650

**Abstract:** Fish biomass and stock assessment are estimated from acoustic volume backscattering strengths (Sv) obtained from various hydroacoustic equipment. Although sound attenuation due to fish schools and water influences the Sv value, only attenuation from water is considered during the acoustic data process. For these reasons, it is necessary to understand the characteristics of sound attenuation by fish. Unfortunately, little is known about sound attenuation from fish. In the present study, the attenuation from one to four fish specimens was precisely measured during ex situ experiments in a water tank. The scientific echo sounder of a split-beam 200 kHz transducer and a miniature hydrophone were used for the attenuation measurements. Results show that the maximum attenuation coefficient ($\alpha$) was approximately 25 dB/cm when the 4 multiple fishes had high fish heights. The relationship between the attenuation coefficient ($\alpha_{200kHz}$) and the total fish height ($H_{total}$) was approximately $\alpha_{200kHz} = 0.54 * H_{total} \pm 0.06$ ($r^2 = 0.72$). This work describes the sound attenuation characteristics to provide basic information for the compensation of Sv from fish schools or layers.

**Keywords:** sound attenuation; attenuation coefficient ($\alpha$); individual and multiple fishes; volume backscattering strength (Sv)





## 1. Introduction

Globally, research has been conducted that continuously and efficiently estimates the biomass of fish using hydroacoustic methods in coastal areas and open oceans [1]. The omnidirectional sonar, multi-beam, and single-beam scientific echo sounders are used to estimate the biomass of fish. In fishery acoustics, the absolute fish biomass is calculated from the volume backscattering strength (Sv), and the accuracy of Sv is important for estimating the biomass of fish [2,3].

Attenuation from fishes increases the natural attenuation from scattering and absorption in echo returns and leads to errors in the estimation of the acoustic biomass using the hydroacoustic method [4]. Attenuation characteristics are determined by the size and shape of the fish, particularly, the presence or absence of swim bladders. An accurate attenuation coefficient ($\alpha$) is required for fish biomass estimation to correct or reduce errors [5,6]. Estimating fish biomass using the theory of echo integration involves the presumption that the linear relationship between the nautical area scattering coefficient ($s_A$) and fish density can be seriously underestimated because of attenuation [1,7,8]. During hydroacoustic surveys, the Sv of the fish in the lower layer is sometimes lower than that of the fish in the upper layer, even within the fish of the same species and size in the water column. Therefore, to estimate the fish biomass accurately, it is necessary to study sound attenuation by fish, to improve the accuracy of fishery echo sounders and the analyses of their data [9,10]. When a scientific echo sounder is used to investigate the biomass of fish based on the current technology, quantitative biomass estimation is possible only when acoustic surveys

are performed. Biomass can be estimated more accurately by compensating $\alpha$, which will be helpful for the quantitative calculation of maximum sustainable yield (MSY) in fisheries [11,12].

Rarely do studies report attenuation of individual fish or schools of fish, such as direct measurement of scattering and attenuation from individual fish [13,14], culture nets [15,16], net cages [17–19], and indirect estimation [20–23]. However, a direct comparison of previous research is difficult because fish species, size, total length, wet weight, attenuation measurement method, and frequency band are different. No studies have examined how the $\alpha$ values differ between individual and multiple fishes.

In this study, we aimed to measure the attenuation characteristics of individual and multiple fishes based on ex situ measurements in a water tank to analyze the attenuation characteristics. From the experimental results, we calculated the relationship between the total fish height ($H_{total}$) and the normalized $\alpha$ at a frequency of 200 kHz. We believe that the results of this study will be significant in understanding the sound attenuation effect for accurately estimating biomass in fishery acoustics.

## 2. Materials and Methods

### 2.1. Attenuation Coefficient

Sound waves disappear in various forms underwater and can be categorized into absorption and scattering [1,24,25]. The extinction cross-section ($\sigma_e$) is generally the effect of an individual finite object on energy removal from the forward-propagating acoustic waves [25]. $\sigma_e$ is the sum of the scattering ($\sigma_s$) and absorption cross-sections ($\sigma_a$) [1,24].

$$\sigma_e = \sigma_s + \sigma_a, \tag{1}$$

where $\sigma_e$ per unit distance in volume in water is denoted by $\alpha$, which is represented by Equation (2) [1,24].

$$\alpha \ (\mathrm{Np/m}) \ = 1/2 \ n \cdot \sigma_e, \tag{2}$$

where $n$ is the density per unit volume of fish and $\alpha$ is the attenuation coefficient per unit distance. In this study, to calculate the attenuation of fish from measurement value, insertion loss (*IL*) with and without fish was incorporated in the experiment [26,27]. *IL* between the transmitter and receiver was calculated from the measured *RL* using Equation (3):

$$IL = RL_{non-fish \ (ref.)} - RL_{fish}, \tag{3}$$

where $RL_{non-fish \ (ref.)}$ is the mean value without the fish (only frame, no fish) and $RL_{fish}$ is the mean value for individual and multiple fishes. The *IL* was converted to $\alpha$ (dB/cm), according to the fish's height (*H*).

### 2.2. Experimental Apparatus

There is a very important precise control of the experimental fish in the acoustic experiments. Actually, it is almost impossible in precise experiments because the coastal ocean contains various currents and flows. For this reason, this experiment was conducted in an indoor freshwater tank without the water flowing.

Ex situ attenuation experiments were conducted in a 5 × 5 × 5 m freshwater tank. A steel frame of 1 m × 1 m × 1 m was placed inside the tank to ensure the accurate arrangement of individual fish. During the experiment, the tank was filled with freshwater at a temperature of 17 °C, and the speed of sound was 1472 m/s [24].

The target fish were Black seabream (*Acanthopagrus schlegelii*), an important commercial species in the coastal ocean of Korea, which swims alone and in schools [28]. Active fish were sampled from a commercial aquaculture farm in the coastal waters of Korea and moved to the experimental indoor freshwater tank in a living state.

In the experiments, the fish was anesthetized using an anesthetization (FA100, 4-allyl-2-methoxyphenol). The 0.5 mL of anesthetization per 1.0 L of seawater was put into and

mixed in a bucket. After the experimental fish were placed in the bucket of anesthetic for about 20 min, the experiment was performed on the anesthetized fish. The experimental fish were physoclistous fishes with closed swim bladders. Therefore, no changes occurred in the size, shape, and gas component of the fish's swim bladder after anesthesia.

A steel frame was added to achieve an accurate horizontal arrangement of the anesthetized fish, tethered using a small monofilament line attached to their mouth and tail. The tethered lines were connected to the steel frame, and vertical lines were attached to a small weight. The tethered depth in each case was normally 1.5–2.1 m. The experimental structure is illustrated in Figure 1.

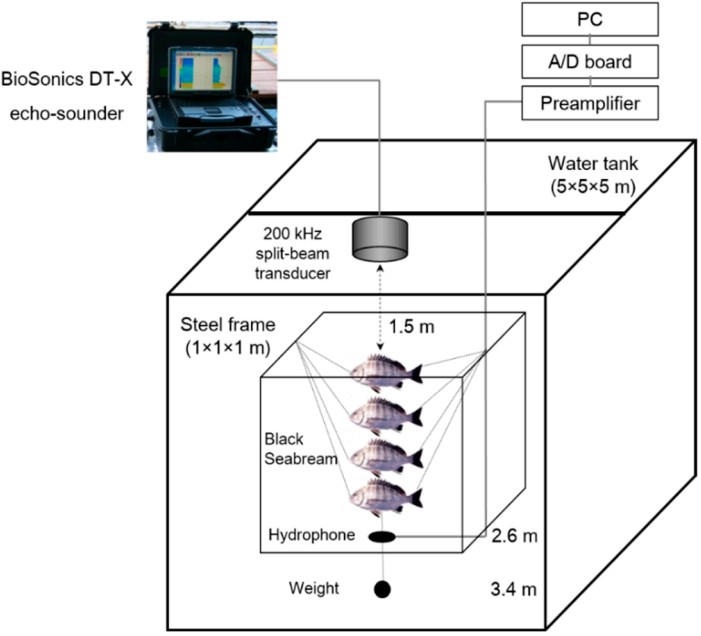

**Figure 1.** Experimental setup for the attenuation measurements of individual and multiple fishes using a scientific echo sounder, along with a 200 kHz split-beam transducer and hydrophone.

A total of 4 individual fish in the ranges of 30.0–34.0 cm, in terms of fork length (*FL*), 10.5–12.0 cm in *H*, and 3.5–5.0 cm in width (*W*) were collected for the attenuation measurements on fish of various sizes and lengths. The details of *FL*, *H*, and *W* for each fish are presented in Table 1 and the experimental cases of arrangement depiction, *FL*, *H*, $H_{total}$, and the number of experiments are listed in Table 2. Sixteen attenuation experiments were performed for ten cases. In the vertical direction, cases 1 to 3 were with 1 fish, cases 4 to 7 with 2 fish, cases 8 to 9 were with 3 fish, and case 10 was with 4 fish. The $\alpha$ for the dorsal aspect was measured after placing the fish in the horizontal direction (tilt angle: the head is 90°) to measure the quantitative $\alpha$. Each fish was arranged in an accurate vertical direction in each experimental case. After the ex situ attenuation experiments, the *FL* and *H* of individual fish were measured, and the specimens were immediately shock-frozen.

**Table 1.** Fork length (*FL*), height (*H*), and width (*W*) of the experimental Black seabream (*Acanthopagrus schlegelii*).

| Black Seabream (Acanthopagrus schlegelii) | | | |
|---|---|---|---|
| **Number** | **Fork Length (FL) (cm)** | **Height (H) (cm)** | **Width (W) (cm)** |
| #1 | 30.0 | 10.5 | 3.5 |
| #2 | 32.0 | 12.0 | 5.0 |
| #3 | 33.0 | 11.0 | 4.5 |
| #4 | 34.0 | 11.5 | 5.0 |

**Table 2.** Details of the experimental cases of the arrangement depiction, fork length, height, total fish height, and the number of experiments.

| | Case 1 | Case 2 | Case 3 | Case 4 | Case 5 | Case 6 | Case 7 | Case 8 | Case 9 | Case 10 |
|---|---|---|---|---|---|---|---|---|---|---|
| Transducer | | | | | | | | | | |
| Arrangement depiction | | | | | | | | | | |
| Hydrophone | | | | | | | | | | |
| Fish number | #1 | #4 | #2 | #1 #3 | #3 #4 | #4 #2 | #2 #4 | #3 #2 #4 | #1 #3 #2 | #1 #3 #2 #4 |
| Fork length [FL] (cm) | 30.0 | 34.0 | 32.0 | 30.0 33.0 | 33.0 34.0 | 34.0 32.0 | 32.0 34.0 | 33.0 32.0 34.0 | 30.0 33.0 32.0 | 30.0 33.0 32.0 34.0 |
| Fish height [H] (cm) | 10.5 | 11.5 | 12.0 | 10.5 11.0 | 11.0 11.5 | 11.5 12.0 | 12.0 11.5 | 11.0 12.0 11.5 | 10.5 11.0 12.0 | 10.5 11.0 12.0 11.5 |
| Total fish height [H_total] (cm) | 10.5 | 11.5 | 12.0 | 21.5 | 22.5 | 23.5 | 23.5 | 34.5 | 33.5 | 45.0 |
| Number of experiments (times) | 1 | 4 | 2 | 1 | 1 | 1 | 1 | 2 | 1 | 2 |

*2.3. Attenuation Measurement and Signal Processing*

The 200 kHz signal, generated using a split-beam transducer of a scientific echo sounder (DT-X; BioSonics, Inc., Seattle, WA, USA), transmitted pulse signals for the ex situ attenuation measurements. Pulse signals that were transmitted through the fish body were received by a hydrophone [6]. The transmitted signals for the pulse duration and ping rate were set to 0.1 ms and 1 ping/s, respectively, considering the distance from the face of the transducer to the fish. The transducer was mounted approximately 0.3 m below the surface and faced in a downward vertical direction. Before any attenuation measurements, the transducer was calibrated using a tungsten carbide sphere and a standard procedure.

The received signals from the attenuating individual and multiple fishes were recorded using a subminiature hydrophone (TC4038, Teledyne RESON; Slangerup, Denmark) and amplified by 30 dB using a voltage preamplifier (EC6081, Teledyne RESON; Slangerup, Denmark). The hydrophone was set approximately 2.6 m below the surface and fixed at the center of the steel frame. The receiving signals were measured for 200 pings in each case. The sampling frequency was 5 MHz, and the direct current (DC) noise was removed from all received signals. A bandpass filter ranging from 0.1 kHz to 1000 kHz was used for the signals. The filtered signals were enveloped using the Hilbert transform, converted into the received level (*RL*) in decibel scales, and averaged. The averaged received signals were compensated for the transmission loss from the range and absorption loss from freshwater. The 200 kHz transducer used in the present study has a very narrow beam width (Table 3). Therefore, the received signals were only affected by experimental fish and the water tank bottom and not the wall. A stabilization time of approximately 10 min was allotted to minimize the effect of the movement on the changes, in the cases before the measurement. The details of the echo sounder and hydrophone settings are listed in Table 3.

**Table 3.** Parameters for the attenuation measurements of individual and multiple fishes using the scientific echo sounder and hydrophone.

| System | Parameters | Value |
|---|---|---|
| Scientific echo sounder at 200 kHz (transmitter) | Frequency (kHz) | 200 |
| | Beam type | Split-beam |
| | Source level (dB @ 1 m) | 221.6 |
| | −3 dB beam width (°) | 6.6 |
| | Absorption coefficient (dB/m) | 0.00962 |
| | Pulse duration (ms) | 0.1 |
| | Ping rate (pps) | 1 |
| Hydrophone (receiver) | Sampling frequency (MHz) | 5 |
| | Gain (dB) | 30 |
| | Receiving Voltage Sensitivity (dB @ 1 V/μPa) | −228 |
| | Low-pass filter (kHz) | 1000 |
| | High-pass filter (kHz) | 0.1 |
| Environments | Water temperature (°C) | 17 |
| | Sound speed (m/s) [5] | 1472 |

## 3. Results

The attenuation of arranged individual and multiple fishes was measured between the transmitter and receiver. A total of 16 experiments were conducted for 10 cases, and 200 pings were transmitted and received in each experiment.

Figure 2 shows a sample target strength (TS) echogram of two fish measurements (case 7) and the attenuation signals collected from the transducer. In the echogram, the TS data from the upper and lower fish were fully separated from the hydrophone, weight, and tank bottom data. The arranged fish in this experiment could be identified because the distance between them was longer than the pulse length. In case 7, the fish heights of the upper and lower were 12.0 cm and 11.5 cm, respectively, with no significant differences (Table 2). However, the difference in the mean TS value was approximately 6 dB. Considering that the value under the same condition does not deviate from that on the center of the beam axis of the transducer, the difference in the TS value could be regarded as the attenuation effect of the fish body and gas-filled swim bladders.

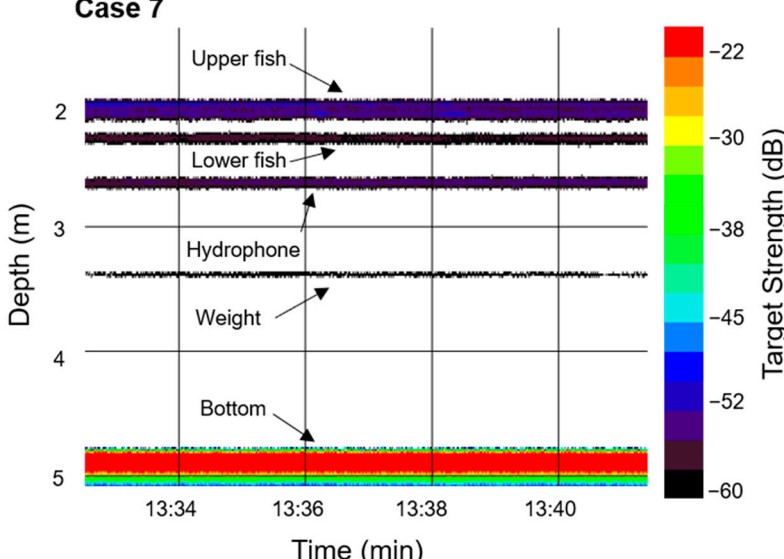

**Figure 2.** A sample of case 7 experiment of target strength (TS) echogram at 200 kHz from multiple fish. The echo signals from the upper and lower fish were fully separated from the subminiature hydrophone, weight, and bottom.

The representative samples, of the received voltage signals and normalized $\alpha$, from no fish, case 1 (one fish), case 4 (two fish), case 9 (three fish), and case 10 (four fish), are shown in Figure 3a and b. It was confirmed that the received voltage signals decreased due to attenuation when the fish were placed between the transmitter and receiver. These results were calculated by averaging the 200 pings of the received signals based on the normalized $\alpha$ in the absence of fish (Figure 3b). The zero-based values of the normalized $\alpha$ were approximately −8 dB, −12 dB, −17 dB, and −25 dB for cases 1, 4, 9, and 10, respectively. These values confirmed that $\alpha$ increased with the $H_{total}$. Based on these values, the normalized $\alpha$ was calculated for all cases. Figure 3c shows the relationship between the $H_{total}$ and the normalized $\alpha$ for all the experimental data. The value of $\alpha$ changed in the same experimental cases. The $\alpha$ varied by up to 5 dB/cm for each individual fish and up to 10 dB/cm for multiple fish. The normalized attenuation coefficient changed between experiments with the same fish individuals because of the alignment of the fish. At this time, the sound waves were transmitted and attenuated by the angle of fluctuation between the fish's body and swim bladder.

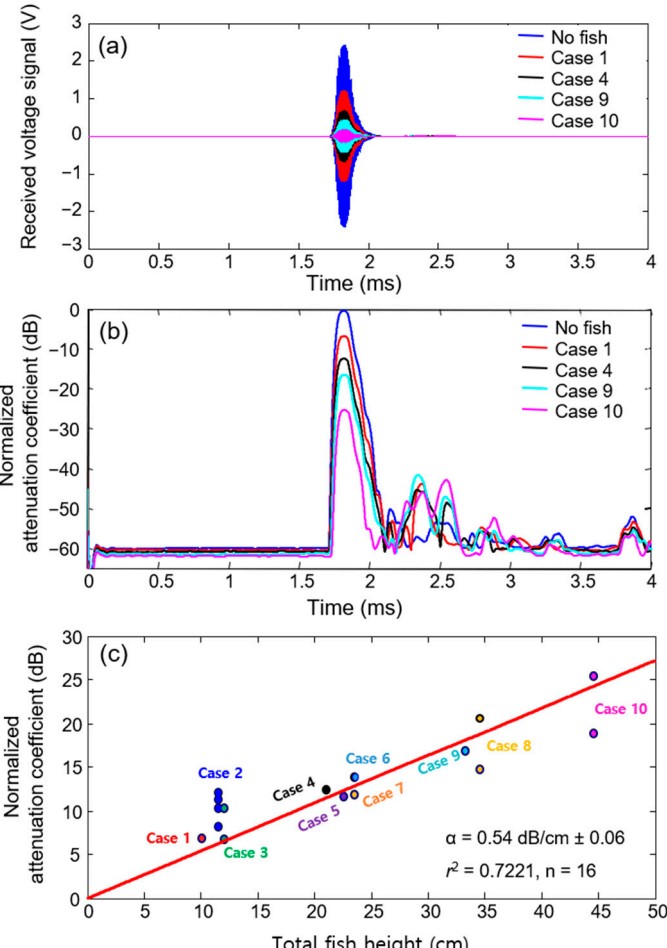

**Figure 3.** (**a**) Received voltage signal and (**b**) normalized attenuation coefficient ($\alpha$) for no fish, case 1, case 4, case 9, and case 10 experiments from the hydrophone, (**c**) the results of the least square linear regression between the total height of the fish and the normalized attenuation coefficient from the experimental all cases.



The normalized $\alpha$ value was calculated by extracting the corresponding peak value based on the transmitting distance. From Equations (2) and (3), using the normalized $\alpha$, the least square linear regression between $\alpha$ and $H_{total}$ was as follows:

$$\alpha_{200kHz} \, (dB/cm) = 0.54 * H_{total} \, \pm \, 0.06 \, \left( r^2 = 0.72 \right) \qquad (4)$$

where $r^2$ is the coefficient of determination. The attenuation effect from individual and multiple fishes tended to decrease as the population increased.

## 4. Discussion

Sound attenuation is not a major problem when locating fish schools underwater using fish finders for fishing activities, yet it is crucial when estimating the quantitative biomass in the coastal ocean. Sv values are key parameters that determine the accuracy of biomass estimation. Assuming that Sv represents the echo integration from individual fish within the sample volume, the numerical density of fish can be estimated by the backscattering cross-section ($\sigma_{bs}$) of individual and multiple fishes [3,17]. Fish attenuation must be elucidated as a correction factor for accurate analysis of the biomass of fish in the ocean.

When the echo integration from the Sv obtained through the scientific echo sounder from the large fish schools or layers is measured, the obtained fish biomass is not accurate because of attenuation. To accurately quantify fish biomass, it is necessary to understand the attenuation characteristics of the fish. Therefore, attenuation must be compensated according to the fish, using a function such as the time varying gain function. This study examined the characteristics of attenuation using individual or multiple fishes as the basis. The results revealed that attenuation must be considered when correcting for Sv. The attenuation effect of fish cannot be neglected when calculating the sound waves passing through the fish, contributing to transmission loss. Attenuation was measured for an important commercial fish, and the $\alpha$ values of individual and multiple fishes are presented from ex situ attenuation measurements.

Since most fish-attenuation studies investigated the attenuation of fish schools in the ocean, a direct comparison with the results of the attenuation of individual fish in this study was not possible. A few studies have shown an indirect relationship between attenuation and the aggregation of several fish species. Davies (1973) performed ex situ attenuation measurements using the direct sphere method for the Northern anchovy at 1–20 kHz and measured the attenuation at a relatively low frequency to observe swim bladder resonance effects [17]. Burczynski et al. (1990) conducted acoustic estimations of Pacific herring in sea pens for attenuation measurements at 420 kHz [21]. Other relevant research studies were analyzed and reviewed by Furusawa et al. (1992). From the results, a normalized extinction cross-section for each frequency was presented through attenuation measurements for fish schools and a comparison with previous research [22]. Diachok (2005) estimated the density based on the attenuation of anchovy schools during day and night at a mid-frequency range of 1–3 kHz, in shallow water environments, through transmission loss measurements [23]. Kim et al. (2021) found that $\alpha$ was approximately 6–15 dB/m from 100–500 fishes in a net cage at 3–7 kHz of the mid-frequency band [19].

In similar studies, such as the scattering of individual fish, Ding (1996) and Ding et al. (1998) measured the scattering, although only the forward/backward scattering values of fish for each frequency and tilt angle were considered; thus, a direct comparison is difficult with this study [13,14]. On the other hand, regarding individual fish attenuation measurements, Freese and Makow (1967) and Sigfusson et al. (2001) measured the attenuation of ultrasound through fish tissues [29,30]. The average $\alpha$ of frozen and thawed whitefish was approximately 0.7 dB/cm at the MHz band [29]. However, this attenuation cannot be compared directly because of the difference in frequency, and only the myomere of the tissue was considered.

TS data were acquired simultaneously to confirm the effect of attenuation in the scientific echo sounder at 200 kHz. TS data from the transducer were examined from the attenuated received data. Figure 4 representatively shows the TS from the case 7 experiment. The 2 fish were arranged vertically, ranging approximately from 1.9 to 2.1 m from the surface of the transducer (Figure 4a and b). The fish position was not distorted when the fish depth was fixed. The TS was measured and compared for two fish of approximately 200 pings each (Figure 4c). At this time, the TS followed a Gaussian distribution and, thus, showed relatively stable mean values (Figure 4d). The upper fish *FL* was approximately 32 cm, and the lower fish *FL* was approximately 34 cm. The upper and lower fish heights were almost the same at 12.0 and 11.5 cm, respectively. In this experiment, two fish were fixed at a distance of approximately 9.0 cm, as confirmed by the depth data of the transducer (Figure 4b). In general, because TS is proportional to *FL*, the TS value of fish located below should be relatively high. However, the TS for the upper fish ranged from −54.7 dB to −50.0 dB (mean TS: −52.0 dB), and for the lower fish ranged from −59.3 dB to −57.3 dB (mean TS: −58.3 dB). The TS value for the lower fish was relatively low at 6–7 dB (Figure 4c). In this result, the TS value should be higher because the *FL* of the lower fish was larger than the upper fish, yet the mean TS value was relatively low, approximately 6.3 dB.

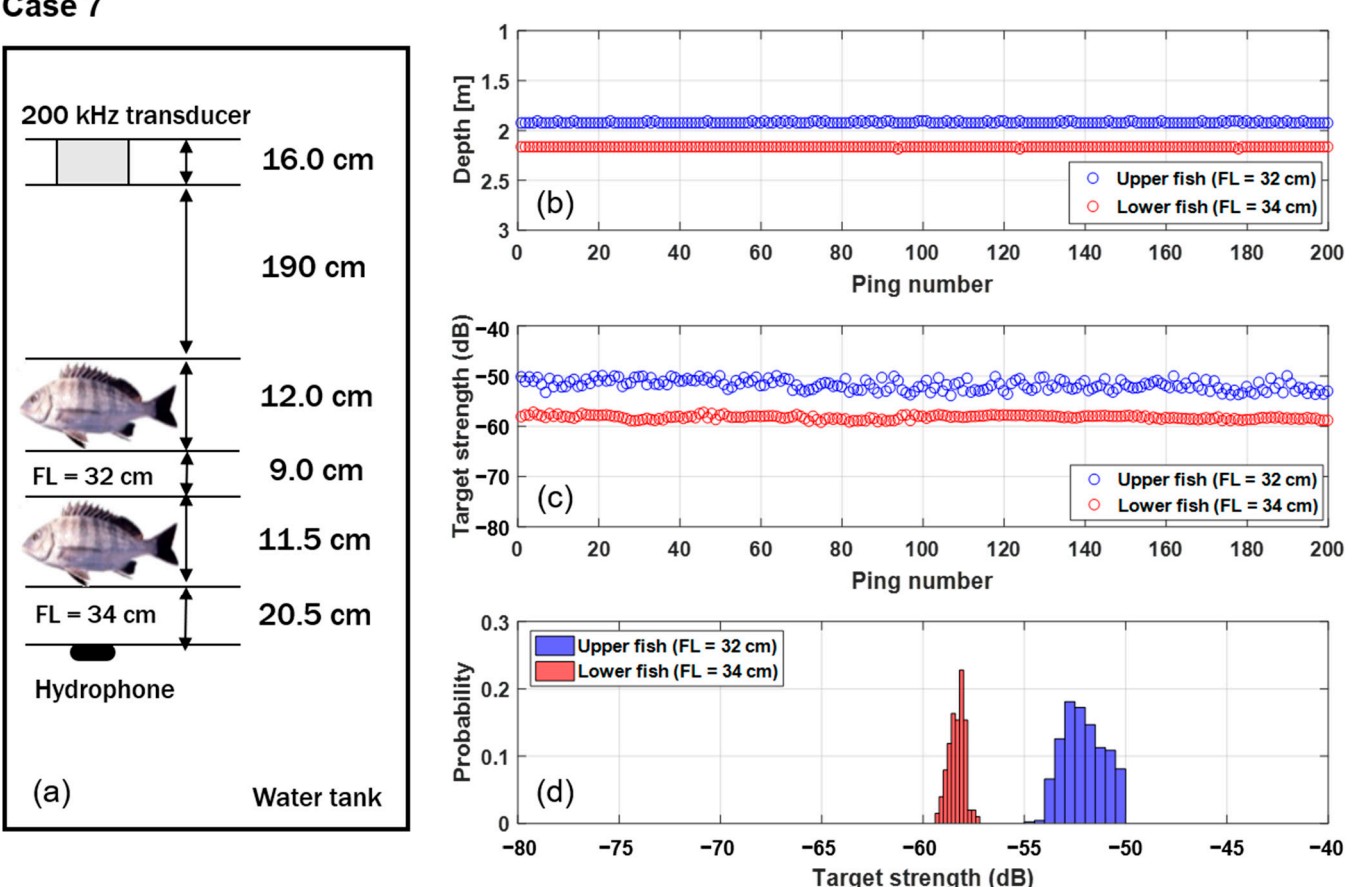

**Figure 4.** Representative attenuation experiments for case 7 from upper fish (fork length, *FL*: 32 cm, fish height, *H*: 12.0 cm) and lower fish (*FL*: 34 cm, *H*: 11.5 cm). (**a**) Experimental setup of the range from the 200 kHz transducer, fish, and hydrophone in the freshwater tank, (**b**) fish depth data, (**c**) target strength (TS) data, and (**d**) the TS histogram from upper and lower fish.

Based on these results, individuals or schools of fish located in the lower layer showed lower TS values than those in the upper layer. In this case, a more accurate and improved estimation of fish biomass should be performed by applying an appropriate fish attenuation coefficient value, according to the thickness of the fish school when measuring TS or Sv.

There are some limitations associated with these experimental results. The fish were artificially placed in the steel frame in a horizontal form, and the attenuation effects may differ depending on individual and multiple fishes. In addition, we tried to match the position of the transmitter, fish, receiver, and the angle of the fish as precisely as possible for each experiment, although some errors may occur due to the alignment of the fish between the transmitter and receiver. Therefore, although some differences occurred in the attenuation coefficient values in the experimental results during the sixteen experiments, the least square linear regression, including the variability, was calculated.

In this study, an experiment was conducted on the acoustic attenuation characteristics of fish to improve the accuracy of fishery resource evaluations using bioacoustics. The results were shown to understand the acoustic attenuation characteristics of fish. However, unfortunately, the experiment was conducted under specific conditions with an indoor freshwater tank. Based on these results, follow-up studies are needed to apply them to future fishery resource evaluations. As a future study, it is necessary to research and quantify fish attenuation characteristics using massive fish populations in actual seawater and native living conditions.

## 5. Conclusions

This study proposes the sound attenuation of individual and multiple fishes based on ex situ measurements. The attenuation coefficient ($\alpha$) was measured using individual and multiple fishes in experiments on ten cases. In this controlled fish-attenuation study, the attenuation characteristics could be confirmed for various fish heights by arranging individual and multiple fishes. In the present study, the normalized $\alpha$ was 0.54 dB/cm $\pm$ 0.06 at 200 kHz. The results provide an understanding of the sound attenuation effect for accurately estimating the biomass of fish. It will also serve as a basis for improving the accuracy of target detection, classification, and localization in naval sonar operations worldwide [21,31,32].

**Author Contributions:** Conceptualization, H.K. and D.K.; Data curation, D.K.; Formal analysis, H.K. and D.K.; Investigation, H.K. and S.C.; Methodology, H.K., S.C. and J.W.C.; Project administration, D.K.; Software, H.K. and S.C.; Resources, J.W.C.; Writing—original draft, H.K. and D.K.; Writing—review & editing, H.K., S.C., J.W.C. and D.K. All authors have read and agreed to the published version of the manuscript.

**Funding:** This research was supported by the Korea Institute of Marine Science & Technology Promotion (KIMST), funded by the Ministry of Oceans and Fisheries, Korea (20210696, Development of Marine Science Exploration Technology in Coastal areas).

**Institutional Review Board Statement:** The study was approved by the Institutional Animal Care and Use Committee at Hanyang University, Seoul, Republic of Korea (protocol code HY-IACUC-23-0009 and 28 February 2023).

**Informed Consent Statement:** Not applicable.

**Data Availability Statement:** The data presented in this study are available on request from the corresponding author. The data are not publicly available due to the data also forming part of an ongoing study.

**Acknowledgments:** We thank Hyungbeen Lee, Su-Uk Son, Joon Seok Park, and Junghun Kim for their help with the attenuation experiments.

**Conflicts of Interest:** The authors declare no conflict of interest.

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
