# Peer review of "Characteristics of Sound Attenuation by Individual and Multiple Fishes"

_fishes, doi:10.3390/fishes8030161_

Round 1

Reviewer 1 Report

The English language needs revision throughout from a native English speaker, preferably an expert in the field who can improve the communication of the intended meaning of the text which in many places is quite unclear.  The document would also benefit from being put through an English spell checker to help remove the spelling mistakes.  A great examples is "Each fish was horizontally arranged in an accurate vertical direction in each experimental case."

Introduction

The acoustics terminology of technical terms such as backscatter, echo, attenuation, interference, impedance etc is used very loosely throughout the manuscript making it difficult to follow the meaning.

"echo-sonder"?

Fish are not "randomly and largely distributed in the ocean" 

Materials and Methods

Again some of the acoustic terminology is imprecisely used.

Doesn't this fish species have a swim bladder?  What happens to the swim bladder when the fish are anaesthetised?

The handling of the experimental fish is not clear - these are marine fish being placed in freshwater, but were anaesthetised - how was this done?  Where is the animal ethics approval?

The experiment was done in a 5x5x5m box - surely there was an enormous amount of reflected and scattered signal from the box?  How was this managed? Presumably isolated pings with only primary signal detected and processed?

The calculations used seawater transmission loss values yet the sampling was done in a tank filled with freshwater.

Results

I am somewhat confused by the number of experiments that were undertaken and how the data from each of these individual experiments is presented in the results. Some of the results seem to be missing, or is it just that the experimental rationale and the corresponding presentation of the results is not well laid out and explained. Furthermore, where multiple experiments were done on a similar set up there are no mean values or estimates of variability of the measurements presented so it is impossible to know the accuracy or precision of any of the measurements undertaken.  There are loose statements made about the angle of the fish swim bladder but this is not measured or reported in the study, indeed the state of the swim bladder for the individual fish is unknown, yet this is likely to be a major determinant of the resulting acoustic measurements.

Discussion

I found the discussion very confusing.  The argument apparently advanced in the introduction in favor of undertaking this research is that it would advance acoustic measurements of biomass of fish schools.  However, the discussion then seems to reveal that the measures for a small number of fish is unlikely to be of much use for such measures, especially given that the attenuation is also likely to be fish and sampling gear specific.  The discussion seems to meander around comparing the results to any other possible results that may be vaguely relevant to this study and does not seem to follow a clearly directed narrative.

Overall this manuscript needs some considerable further work to bring it up to a standard where it is suitable for publication.

Author Response

We have attached a Response to the reviewer`s comments.

Reviewer 2 Report

Review of: ‘Characteristics of sound attenuation by individual and multiple fishes’

This paper presents results of experiments to study attenuation of sound in fish, in this case specimens of black seabream.  The sound source used in the experiment was a 200 kHz split beam echosounder.  The study was done as a controlled experiment in a water tank where the fish were suspended in a vertical aspect with respect to the sound propagation direction, ie the sound penetrated from the dorsal or top of the fish.  Multiple experiments were carried out with single and multiple (up to four) fish in the same aspect.   Attenuation was inferred with reference to the signal strength in the absence of fish.  The main feature of the experiment was the measurement of the sound transmitted through and the sound echo from the fish.   This provides the means to obtain an estimate of the attenuation due to the fish and at the same time, a measure of the echo return from the fish.   Thus, the quantitative impact of attenuation loss in the fish could be inferred as it affects the target strength determined from the echo from the fish or array of fish. 

However, although this impact is mentioned, the authors do not attempt to use the results of their experiment to address this question.  For instance, in the case presented in figure 4 for target strength, the results indicate a difference of about 6-7 dB for TS.  Can this difference in TS accounted for quantitatively from the estimates of attenuation in the upper of the two fish?   If I use the measured attenuation at 200 kHz of 0.54 dB/cm, I get about 6 dB for the loss in propagating through the fish.  I realize that the results of the ex situ study cannot translate directly to the open ocean case, there should be some attempt to quantify the impact of attenuation in the controlled experiment.  I see this as a shortcoming of the paper.

Overall, the paper is generally well written, except for some statements that make no sense at all.  There are also many instances of typos or incorrect grammar, far too many to list separately.  Instead, an annotated version of the paper is attached with all the comments.

The experiment was carefully done and the results are reasonable and should be reported after some revision to clarify and improve the message.  My recommendation is that the authors should address the questions, comments and errors identified in the annotated version, and at least make some attempt to quantify the impact of attenuation in their study.

Most of my comments are in the attached annotated file.  I will mention a few of the more critical ones here.

1. P1 Introduction:  sentences ‘For example…..absorption.’ These three sentences are confusing and disorganized; I suggest the following revision.   I think what you are trying to say is that attenuation in the fish needs to be taken into account to obtain reliable and accurate estimates of fish biomass based on measurements using echosounder data.  The third sentence (Large concentrations...) should be revised and put first of the three: something like 'Attenuation from marine organisms increases the natural attenuation from scattering and absorption in echo returns.'  Then use the example of the Sv results from targets at different depths.  Then state the hypothesis that attenuation from the marine organisms can lead to incorrect estimates of the fish biomass if not correctly taken into account.

2.  Page 2: Was there a reason to use fresh water?  I assume the sea bream are salt water fish?  I realize that the type of water may not affect the measured value of fish attenuation, but readers may ask this question.

3.  Page 6: ‘This finding shows…bladder.’  I do not understand this statement at all.  It seems to me that your results show that (1) the attenuation increases with the number of fish (which in your case is determined by the parameter H) and (2) the specific orientation of the fish (ie relatively minor tilts in the fish from the vertical aspect) affects the measured  value of attenuation.  You should revise the highlighted statement accordingly.

4.  P7: comparison with other measured values of attenuation:  Does the value from your experiment for attenuation at 200 kHz appear to be consistent with the value at higher and lower frequencies?  I realize that comparison is difficult because of the different experimental conditions, but it would be useful to make a stronger statement to set your value in context of other known values.

5.  P8: ‘Calculation of the attenuation is necessary…fish biomass.’  This is very weak and confusing statement to make.  Instead, it would be very useful for the authors to make some attempt to show how the fish attenuation affects the different TS measurements in their study. 

6.  P9: Conclusion: Also need to point out that the results here are limited to the frequency used in the experiment.  Can any comment be made about the frequency dependence of the attenuation>

Author Response

(The authors gave the same response as above.)

Round 2

Reviewer 1 Report

This study was conducted on a vertebrate without any animal ethics oversight or approval.  It involved transferring living fish into a highly stressful environment, in which the fish would have suffered if not appropriately managed. The management framework for the fish is not outlined in the study, so it is not possible for me to assess whether it may have been effective or not. As it stands this manuscript is not consistent with MDPI animal welfare policy; “If no animal ethics committee is available to review applications, authors should be aware that the ethics of their research will be evaluated by reviewers and editors. Authors should provide a statement justifying the work from an ethical perspective, using the same utilitarian framework that is used by ethics committees.

It is a requirement for MDPI journals that the manuscript clearly states how steps were taken to ensure the fish did not suffer during experimentation. As it stands the manuscript provides insufficient details on anaesthesia methods, and how sufficient anaesthesia was maintained within the experimental set up for the duration of the study (>10 minutes), while the fish would have been undergoing massive physiological trauma.

The manuscript has been improved by the authors. It still remains difficult to connect all the experiments presented in Table 2 with any subsequent systematic reporting of the results for these series of experiments. Indeed it appears as if much of the data that was collected was not analysed and presented - this makes it very confusing for the reader.  The manuscript needs much clearer connection between the methods and the results presented, rather than just selecting specific results out of the full set of experiments undertaken and presenting them as representing the outcome of all the research undertaken.

Author Response

We attached the responses to the reviewer`s comments.

Reviewer 2 Report

I am satisfied with the responses from the authors.

Author Response

Thank you for your comments and suggestion. 

Round 3

Reviewer 1 Report

The author's have resolved the animal ethics concerns of the reviewer by providing an institutional approval, and some detail of the method of anesthesia applied.  It would be useful to include a reference for the anesthetic method that was applied and the depth of anesthesia maintained through the method and how fish were euthanised at the completion of the experiment, which I assume was also included in the animal ethics approval.  This is fundamental requirements for studies involving vertebrates.

In my view the authors have failed to inadequately address the disconnect between the experimental design and the set of results presented.  It appears that a large set of experiments were undertaken, yet only selected results are presented.  It is therefore somewhat superfluous, as well as confusing for the reader, to have the additional research methods presented when they are irrelevant to the results presented and discussed in the manuscript.  The reluctance of the authors to sort this matter out is frustrating, but if they did it would make for a much improved manuscript that would be a great deal easier for readers to understand, and of more value to the wider scientific community.

Author Response

We have attached the revised manuscript and Responses to the reviewer`s 1 comment.
